# The Relationship of Tobacco, Alcohol, and Betel Quid with the Formation of Oral Potentially Malignant Disorders: A Community-Based Study from Northeastern Thailand

**DOI:** 10.3390/ijerph18168738

**Published:** 2021-08-19

**Authors:** Prangtip Worakhajit, Pornpoj Fuangtharnthip, Siribang-on Piboonniyom Khovidhunkit, Pim Chiewwit, Boworn Klongnoi

**Affiliations:** 1Department of Oral and Maxillofacial Surgery, Faculty of Dentistry, Mahidol University, Bangkok 10400, Thailand; prangtip.wor@mahidol.ac.th (P.W.); pim.chw@mahidol.ac.th (P.C.); 2Department of Advanced General Dentistry, Faculty of Dentistry, Mahidol University, Bangkok 10400, Thailand; pornpoj.fun@mahidol.ac.th (P.F.); siribangon.pib@mahidol.edu (S.-o.P.K.)

**Keywords:** risk factors, tobacco, alcohol, betel quid, oral potentially malignant disorders (OPMDs), oral cancer screening

## Abstract

This study’s objective was to describe the relationship between the main risk factors for oral cancer, including tobacco (in the form of cigarettes, smokeless tobacco (SLT), secondhand smoking (SS)), alcohol, and betel quid (BQ), and the occurrence of oral potentially malignant disorders (OPMDs). A community-based case-control study was conducted with a population of 1448 adults aged 40 years or above in northeastern Thailand. Patients aged 60 years or above (OR 1.79, *p* < 0.001) and female patients (OR 2.17, *p* < 0.001) had a significant chance of having OPMDs. Our multivariate analysis showed that the most potent risk factor for OPMDs occurrence was betel quid (BQ) (adjusted OR 4.65, *p* < 0.001), followed by alcohol (OR 3.40, *p* < 0.001). Even former users were at risk of developing OPMDs. The synergistic effect between these main risk factors was significantly shown in the group exposed to SLT, SS, BQ, and alcohol. The most potent synergistic effect was found in the group exposed to SLT, BQ and alcohol with the OR = 20.96.

## 1. Introduction

Oral cancer was ranked 16th among the 36 cancers in 2018. It was one of the leading causes of death worldwide, with 177,384 deaths and an estimated 354,864 new cases [1].

A study conducted in Thailand in 2013 found that the female population in the area chew betel nuts, smoke, and drink alcohol. In this study, Betel quid (BQ) chewing was a significant risk factor for the development of oral cancer within that geographical region [2].

Oral potentially malignant disorders (OPMDs) are widely known for their potential of transformation to oral cancer [3]. Moreover, many studies have revealed that the risk factors for OPMDs are similar to those for oral cancer (oral squamous cell carcinoma) [4]. The likelihood of OPMDs transforming into oral cancer depends on many factors, including the location of the lesion, its clinical characteristics and size, the age of the patient, the duration of the onset of the lesion, and the patient’s behaviors [5]. Although OPMDs are less commonly found in females [4,6,7,8], the Indian study by Silverman et al. in 1976 showed that the female population had a higher malignant transformation rate compared to males [9]. This requires gender factors to be further considered in the development of lesions. A 2014 study by Wang et al. found that the overall rate of malignant transformation was 4.3% [10]. In addition, the malignant change was significantly greater for lesions characterized by mucosal dysplasia [5]. It was found that OPMDs on the tongue and floor of mouth were less likely to be found but had a greater chance of turning into malignancies [5,11,12,13,14,15]. Hence, understanding the risk factors for both conditions is crucial and plays an important role in oral cancer screening and prevention. According to the population-based study of an oral cancer screening program in Taiwan, delayed diagnosis and mortality were reduced by 21% and 26 %, respectively. It also found a 45% greater survival rate in the screened group compared to the unscreened group [16].

Concerning risk factors for both conditions, the use of tobacco and tobacco products (e.g., snuff) and betel quid (BQ) chewing are widely accepted as very potent risk factors, and alcohol drinking is also a commonly associated risk factor [17,18,19]. Previous studies have shown an association between each individual risk factor and OPMDs or oral cancer [4]. Nevertheless, studies of the synergistic effects among these common risk factors are still limited. A statistically significant association between OPMDs and habits has been demonstrated in many studies, although regional differences exist [20,21]. In Asia, leukoplakia is known to be associated with BQ (pan, areca quid) chewing and smoking (bidi, cigarette) [22], whereas in Western countries, it is associated with cigarette smoking, snuff, and alcohol consumption [21]. Therefore, this study was performed to analyze the relationship between these common risk factors and the occurrence of OPMDs as well as their synergistic effects when patients are exposed to more than one risk factor.

This community-based case-control study was undertaken as part of an oral cancer screening project in Thailand Health Region 9 (the provinces of Nakhon Ratchasima, Chaiyaphum, Buriram, and Surin.). The aim of the study was to evaluate the relationship between the main risk factors for oral cancer, including tobacco (in the form of a cigarette, smokeless tobacco (SLT), secondhand smoking (SS)), alcohol, and betel quid (BQ), as well as their synergistic effects, and the occurrence of oral potentially malignant disorders (OPMDs).

## 2. Materials and Methods

Based on a review of the literature and related research by Kumar et al. (2015) [23], the incidence of OPMD lesions was 28.4% in smokers and 8.4% in non-smokers. The sample size was calculated using the formula for binary logistic regression.
(1)$$n={\left[\frac{Z_{1-\alpha /2}\sqrt{\bar{p}\bar{q}}\left(1+\frac{1}{r}\right)+Z_{1-\beta }\sqrt{p_{1}q_{1}+\frac{p_{2}q_{2}}{r}}}{\Delta }\right]}^{2};$$

n refers to the sample size. $p_{1}$ and $p_{2}$ are the prevalence of exposed and non-exposed group, which equal 0.284 and 0.084, respectively. $Z_{1-\alpha /2}=1.96$ for 95% confidence interval and $Z_{1-\beta }=1.28$ when $\beta =0.01.\;\bar{p}=$$\frac{p_{1}+p_{2}r}{1+r}$, and $\bar{q}=1-\bar{p}$ when r=1, in this case $\bar{p}=0.184$ and $\bar{q}=0.816.\;\Delta$ is the difference between $p_{1}$ and $p_{2}$ = 0.2.

In order to reduce data discrepancies, the sample size was compensated by increasing the size of each group by 10%. Therefore, the sample size was 85 samples for each group, and the final result was calculated to be at least 170 samples.

The calculated n was the least number that could be used to create statistical results. Nevertheless, as a part of the leading research project, the Development of Disease Management Model of Oral Cancer with an Integration Network of Screening, Surveillance, and Treatment in Nakorn-Chai-Bu-Rin project, involved a wide-ranging study area; there was inevitably a large number of participating populations. This study, therefore, decided to collect patient data with completeness up to the community hospital level, in which specialized dentists diagnosed the lesions. This judgment met the inclusion and exclusion criteria of the research.

Data collection began with an initial screening at the village level (S1) with an oral cancer risk screening questionnaire administered by healthcare volunteers. Subsequently, patients with risk factors were referred to the residential sub-district hospital (S2) for further examination by an oral hygienist. The oral hygienists screened for lesions and recorded all details of the patients’ risk factor exposure histories. Based on the sub-district screening, data were recorded in the online research operating system database for the oral cancer project. Screening continued at the community hospital (S3), where dentists cooperated with the research team from Mahidol University and the Center of Excellence in Oral Cancer Maharat Nakhon Ratchasima Hospital, working together for the oral examination, diagnosis, and treatment. According to the S3 level screening form, it was possible to identify vulnerable patients with OPMDs by providing diagnoses based on clinical features.

The data were obtained from August 2019 to February 2021. A total of 1448 patients aged 40 years old or older were enrolled in this study. Inclusion criteria were the data from the leading research project, screening by specialized dentists at the S3 level. Exclusion criteria were data from patients who had a history of head and neck cancer, patients who used medications or had systemic conditions related to the abnormal oral manifestation, and data from dropout patients or incomplete records. The Ethics Committee approved the study of the Faculty of Dentistry/Faculty of Pharmacy, Mahidol University, and Institutional Review Board (Approval code COA.No.MU-DT/PY-IRB 2021/021.1702 and date of approval 17 February 2021).

Smokers were defined as those who had smoked at least one cigarette per day for six months continuously. Frequency was recorded as cigarettes per week, and this included the duration of the habit. Alcohol use was defined as drinking alcoholic beverages at least once a week continuously, including beer, hard liquor, and herbal liquor, whereas wine was uncommon in the study area. BQ chewing was defined as chewing BQ for at least six months continuously. The dropouts were found at the junction of each level of screening. The number of participants and dropouts were presented in Figure 1.

The data were analyzed using the IBM SPSS Statistics for Windows, version 25.0 (IBM Corp., Armonk, NY, USA). The baseline characteristics were analyzed with descriptive statistics by frequency and percentage in categorical data, and mean, standard deviation, median and interquartile range were used in continuous data. Continuous variables were compared using the independent T-test or Mann–Whitney U test, and categorical variables were compared using the Chi-squared test or Fisher’s exact test, as appropriate. The risk factors associated with OPMDs were selected in the logistic regression analysis performed by univariate and multivariate analysis. The odds ratio (95% CI) was presented with a *p*-value < 0.05 was considered statistically significant. Kaplan–Meier survival analysis, with a log rank test, was used to compare OPMDs presentation related to duration of exposure, between each risk factor.

## 3. Results

Overall, 72.7% of patients were exposed to or had a history of exposure to one or more main risk factors ((Table 1), (Figure 2)), including 14.6% smokers, 5.5% former smokers, 18.9% SS, 6.4% SLT users, 1.0% former SLT users, 36.3% BQ chewers, 2.4% former chewers, 16.3% alcohol drinkers, and 6.8% former drinkers. Our study had more female participants (N = 992) than males (N = 456). As the results also showed that there were differences in risk factor exposure among different gender. Smoking and drinking habits were mainly found in male patients, whereas most female patients exposed to tobacco smoke were exposed to SS. Female patients used SLT and chewed BQ more commonly than male patients.

A univariate analysis (Table 2) showed that OPMDs were more significantly found in females (2.17 times more than that in males). Patients aged 60 years or older had a 1.7-fold greater risk of lesions compared to the younger group. Exposure to one, two, or three risk factors increased the occurrence of lesions 3.04-, 5.40-, and 24.82-fold, respectively. In the tobacco group, there were no statistically significant results in current smokers. However, in the group of former smokers, the result showed the statistical association to OPMDs occurrence (OR 0.26 [0.14, 0.49], *p* < 0.001). In SLT users, there was a significant result in current users who were at risk of having OPMDs (OR 3.98 [2.52, 6.30], *p* < 0.001). In the BQ group, the results showed that not only current chewers (OR 6.91 [5.43,8.79], *p* < 0.001) but also former chewers (OR 6.89 [3.37, 14.10], *p* < 0.001) had a strong association to OPMDs occurrence. Those who chewed BQ for more than 30 years were 1.88 times more likely to have lesions than those who chewed for less than 30 years (OR 1.88 [1.31, 2.72], *p* = 0.001). For the alcohol drinkers, the risk OPMDs was significantly associated with the group of current drinkers (OR 1.49 [1.12, 1.96], *p* = 0.007). All univariate analysis between the single risk factors exposure and OPMDs occurrence is described in Table 3.

A variable with a *p*-value of less than 0.2 was then selected for the multivariate analysis (Table 4). It was found that gender was statistically associated with the occurrence of OPMDs (adjusted OR 1.64 [1.10, 2.43], *p* = 0.015). BQ chewing as well as alcohol drinking were significantly associated with the occurrence of OPMDs with adjusted OR of 4.65 ([3.29, 6.58], *p* < 0.001) and 3.40 ([2.23, 5.18], *p* < 0.001), respectively.

The synergistic effect among risk factors was raised as an issue in our study. An analysis of the group exposed to more than one factor was performed (Table 5). We could find the significant synergistic effect from the combination of each risk factor but not in the tobacco group. The combination of SLT + alcohol and SLT + BQ resulted in the synergistic effect of 13.79-fold and 4.65-fold, respectively. Current BQ chewers with alcohol drinking showed an increased chance of developing a lesion of 9.33-fold, and for BQ chewers with SS, 2.97-fold. For a person exposed to SS who also consumed alcohol, the occurrence of lesions was increased to 3.41-fold. For the three risk factors exposure, a group analysis was also described in this study. Significant results were found in the group of BQ chewers with alcohol in combination with SLT or SS at an increased ratio of 20.96 and 7.30-fold, respectively.

The cumulative effect ((Table 6), (Figure 3)) of each risk factor was compared using the%age of OPMDs presentation at each interval of exposure (10-, 30-, and 50-year). Among the smoker group, OPMDs were found to be 7.5%, 28.2%, and 57.9% for the durations of 10, 30, and 50 years, respectively. The SLT group resulted in the occurrence of OPMDs of 16.1%, 50.6%, and 82.9% for 10-, 30-, and 50-year intervals. BQ chewing was likely to cause OPMDs of 12.3%, 44.7%, and 83.6% for 10-, 30-, and 50-year intervals. Alcohol drinking ranked from 20.3%, 49.8%, and 78.5% for 10-, 30-, and 50-year intervals.All risk factors caused OPMDs with statistically significant results (log-rank = 36.54, *p* < 0.001).

## 4. Discussion

Our studies show that age affects the incidence of OPMDs. A significant difference was found between the groups under 60 years of age and 60 years or older. In the group aged 60 years or above, there was a 1.79-fold higher rate of OPMD occurrence. The finding of lesions among the elderly is consistent with past studies [24,25]. In the 1977 study by Bánóczy et al. in Hungary [12], it was found that the peak incidence of leukoplakia was in the sixth decade, but the highest rates of transformation were in the seventh decade (7.1%) or in patients over 71 (8.2%). Large studies from India and the developing world support the view that lesions are more likely to develop and progress in older individuals [6,7,26,27].

Our study found that female patients had 2.17 times more lesion presentation than males, unlike the past study by Lind et al. in 1987 in the Norwegian population. They found lesions in 102 males and 55 females, while oral cancer developed in eight males (7.8%) and six females (10.9%). Studies in large populations [6,7,8] found that females were much less likely to have lesions. A study of Thailand by Anchisa et al. in 2019 [28] also found that the ratio of males with lesions was greater. It is important to realize that past studies also reported that there was a higher chance of malignancy transformation in female patients than males [5]. The proactive fieldwork of our research team has allowed us to reach the population more thoroughly than in the past. Many populations have entered the screening system, making it possible to find lesions in groups normally missed by other screening programs.

The favored form of tobacco use can vary across geographic areas and cultures. Cigarettes, cigars, and pipes are the major types of tobacco smoking, while chewing tobacco and snuff are the most common forms of SLT. Smoking has long been implicated in the etiology of oral cancer and OPMDs [4], and many studies [6,16,19,21,22,23,25,26,29,30,31,32,33,34,35,36] have shown a positive association between smoking and OPMDs. In our study, conducted in northeastern Thailand, cigarette smokers were commonly found compared to the SLT users at a ratio of approximately 3:1. SLT use has been implicated as a risk factor for both oral leukoplakia and oral cancer [37,38,39,40,41,42]. Furthermore, in our study, there was a statistically significant relationship between current SLT use and the occurrence of OPMDs (3.98-fold). According to a study from Puerto Rico in 2011 [43], tobacco smoking was strongly associated with the risk of OPMDs. There was a more than four-fold increased risk among the current versus the never-exposed group; however, the risk was notably attenuated among former smokers. These findings are consistent with numerous previous studies of OPMDs [6,16,19,21,22,23,25,26,29,30,31,32,33,34,35,36], and oral cancer [44,45,46,47,48]. In our study, we found that the duration and number of cigarettes smoked increased the risk of OPMDs, but these were not statistically significant. However, the one who quit smoking could reduce the chance of OPMDs occurrence compared with a current smoker.

Alcohol is a risk factor for many cancers, including cancers of the oral cavity [47,48,49,50]. The type of alcoholic beverage and frequency of consumption have an effect on cancer risk, and the risk is increased when alcohol is used with tobacco products [47]. Oral cancer risk is likely related to overall alcohol consumption (number of years drinking) rather than the amount of drinking per day [47]. Alcohol is strongly associated with the development of oral cancer. It has also been proven that the prolonged use of alcohol can lead to atrophy of the oral mucosa, as well as stimulation of excessive mucosal chemistry leading to greater susceptibility to carcinogens (and alcohol itself is a carcinogen) [49], although the association with OPMDs is currently unclear [4]. Alcoholic beverages consumed by our population included both hard liquor and beer in combination. In contrast to many past studies [21,23,36,51,52], our results showed an association between alcohol drinking and OPMD occurrence. Other similar results reported a positive association with OPMDs [19,25,32,37,43,53,54,55,56,57].

BQ chewing is the most common behavior in Southeast Asia. The BQ contains betel leaf, betel nut, and khaini. In Thailand, turmeric is generally added. Another important component of betel is burnt tobacco, boiled tobacco, or tobacco and molasses. In 2012, in northeastern Thailand, the chewing rate of BQ was high, especially among elderly women [58]. Several studies have shown that areca nut and betel ingredients are associated with the development of oral cancer lesions [2,16,22,42]. Several studies have addressed betel’s carcinogenicity; BQ contains substances that cause genetic changes. In an in vitro study with fibroblasts from the oral epithelium, it was shown that the key components of BQ exhibited genotoxicity, cytotoxicity, and cell division properties related to oral cancer pathophysiology [59]. A high risk of developing OPMDs was observed with daily chewing of BQ in our study, similar to many previous studies [19,23,24,25,29,60,61]. Moreover, our results showed that patients who chew BQ for 30 years or more have a 1.88-fold chance of developing lesions compared to the control group.

From our study found, it was found that the exposure to one, two, or three risk factors increased the occurrence of lesions 3.04-, 5.40-, and 24.82-fold, respectively. This result can be implied by our hypothesis that there was a possible tendency of having a synergistic effect among risk factors. Adding tobacco to BQ is a common practice in Southeast Asian countries [23,62,63]. For current smokers, there was no significant effect on OPMDs. However, when smoking was combined with BQ chewing, it was found to be a significant association. Our study also found that SLT use was significantly associated with the development of lesions. Moreover, if SLT was combined with other risk factors, it was found to have a significant association with OPMDs. The use of SLT and BQ chewing increased the occurrence of OPMDs 4.48-fold, and when combined with alcohol consumption, there was an 8.56-fold greater risk compared to the group using only SLT. Moreover, when SLT was used with BQ chewing and alcohol drinking, there was a significantly higher trend of having OPMDs of 20.96-fold. This was in line with a study by Amarasinghe et al. in Sri Lanka in 2010 [29]. The smoking and drinking interaction was mentioned in many past studies, highlighting that it had an association with the occurrence of oral cancer and OPMDs [23,35,44]. Our analyses revealed no evidence that alcohol consumption modified the effect of smoking in terms of OPMD risk. Concurrently, in the group exposed to tobacco, BQ and alcohol presented an absence of OR due to limitations of population; however, all patients in this group presented OPMDs.

Interestingly, our study found that the SS group alone did not have any association with disease, but when combined with BQ chewing and/or alcohol drinking, they were significantly prone to OPMDs. This issue has not been mentioned in previous studies; therefore, it should be incorporated into health promotion programs for oral cancer prevention.

The limitation of our study was the distribution of the risk factors among our population. We found that the most extensive risk factor was betel quid. Concerning tobacco products, our study showed different consumption between genders. The male population mainly smoked cigarettes but rarely used SLT. In contrast to females, they used much more SLT than cigarettes. These behaviors are partly found in some Asian countries but do not normally exist in other continents, where the use of betel quid and SLT are not ubiquitous.

Using the community network in combination with multilevel dental care is a highly effective model. After proving its effectiveness, this model will be implicated in another health region’s future researches and could be used as a national public health policy for oral cancer screening in Thailand or elsewhere. This study proved a strong association of the main risk factors to the occurrence of OPMDs in all exposure characteristics, especially in the case of those exposed to combined risk factors. The results of this study might essentially draw healthcare practitioners’ attention to helping their patients avoid or stop risky behaviors. This study is part of a leading research project that is the largest proactive oral cancer screening project conducted in Thailand to date.

## 5. Conclusions

Direct exposure to tobacco products, BQ, and alcohol is associated with the occurrence of OPMDs. It was found that the presentation of OPMDs was closely associated with BQ chewing and alcohol consumption. The synergistic effect of common risk factors is well demonstrated in this study. The association of risk factors as well as duration of exposure reported in this study can be used as clues to find OPMD lesions in routine oral examinations by dentists and health care workers.

## Figures and Tables

**Figure 1 ijerph-18-08738-f001:**
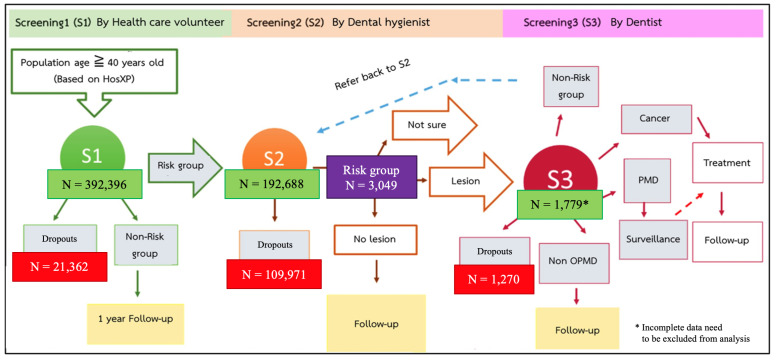
Diagram of project workflow.

**Figure 2 ijerph-18-08738-f002:**
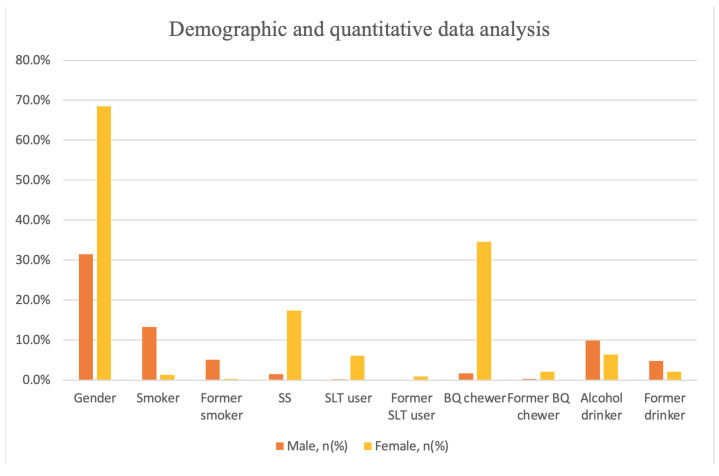
Demographic and quantitative data analysis.

**Figure 3 ijerph-18-08738-f003:**
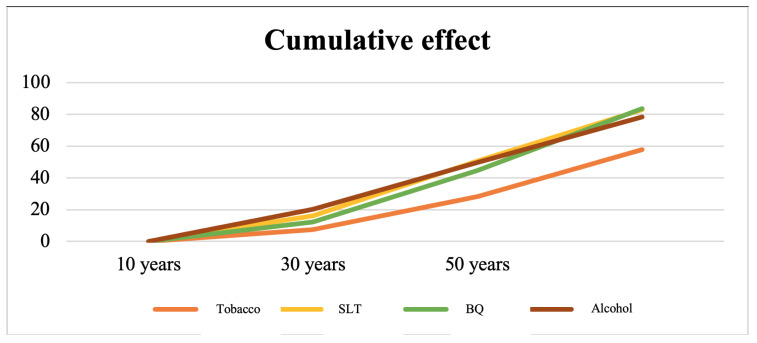
Cumulative effect of duration of exposure to the occurrence of OPMDs.

**Table 1 ijerph-18-08738-t001:** Demographic and quantitative data analysis.

	Male, n%	Female, n%	Total
**Gender, n%**	456 (31.5)	992 (68.5)	1448
**Age (year), mean ± SD**	65.06	66.93	66.34 ± 10.15
**Tobacco, n%**			
Smoker	192 (13.3)	19 (1.3)	211 (14.6)
Former smoker	74 (5.1)	5 (0.3)	79 (5.5)
SS	22 (1.5)	252 (17.4)	274 (18.9)
**SLT, n%**			
User	3 (0.2)	89 (6.1)	92 (6.4)
Former user	2 (0.1)	13 (0.9)	15 (1.0)
**BQ, n%**			
Chewer	25 (1.7)	501 (34.6)	526 (36.3)
Former chewer	4 (0.3)	31 (2.1)	35 (2.4)
**Alcohol, n%**			
Drinker	143 (9.9)	93 (6.4)	236 (16.3)
Former drinker	69 (4.8)	30 (2.1)	99 (6.8)

SS, secondhand smoker; SLT, smokeless tobacco; BQ, betel quid. The baseline characteristics were analyzed with descriptive statistics by percentage in categorical data, and mean, standard deviation were used in continuous data.

**Table 2 ijerph-18-08738-t002:** Odds ratios of personal data and number of risk factors with OPMDs occurrence.

Categories	Odds Ratio	(95% CI)	*p*-Value
**Gender**			
**Male**	1.00		
**Female**	2.17	1.65, 2.85	<0.001 *
**Age**			
<60 years	1.00		
≥60 years	1.79	1.38, 2.32	<0.001 *
**Risk Factors**			
**Number of exposed risk factor(s)**			
None	1.00		
1 risk factor	3.04	2.31, 4.00	<0.001 *
2 risk factors	5.40	3.91, 7.44	<0.001 *
3 risk factors	24.82	10.16, 60.61	<0.001 *

Data were analyzed with simple logistic regression. * Significant level at *p* < 0.05.

**Table 3 ijerph-18-08738-t003:** Odds ratios of risk factor exposure characteristic with OPMDs occurrence.

Risk Factors	Odds Ratio	(95% CI)	*p*-Value
**Tobacco**			
**Current**	0.89	0.66, 1.21	0.467
**Timing of tobacco smoking**			
<30 years	1.00		
≥30 years	1.41	0.78, 2.53	0.258
**Smoking frequency per week**			
<35 cigarettes	1.00		
≥35 cigarettes	1.27	0.72, 2.25	0.406
**Former**	0.26	0.14, 0.49	<0.001 *
**Period of quit smoking**			
<10 years	1.00		
≥10 years	0.41	0.11, 1.51	0.179
**Period of past smoking**			
<20 years	1.00		
≥20 years	0.56	0.16, 2.03	0.380
**SS**	0.99	0.76, 1.30	0.962
**SLT**			
**Current**	3.98	2.52, 6.30	< 0.001 *
**Timing of using**			
<25 years	1.00		
≥25 years	1.07	0.44, 2.60	0.887
**Former**	2.61	0.93, 7.39	0.070
**Period of quit using**			
<10 years	1.00		
≥10 years	0.40	0.05, 3.42	0.403
**Period of past using**			
<20 years	1.00		
≥20 years	0.63	0.07, 5.35	0.668
**BQ**			
**Current**	6.91	5.43, 8.79	< 0.001 *
**Timing of chewing**			
<30 years	1.00		
≥30 years	1.88	1.31, 2.72	0.001 *
**Former**	6.89	3.37, 14.10	<0.001 *
**Period of quit chewing**			
<5 years	1.00		
≥5 years	0.94	0.22, 3.92	0.930
**Period of past chewing**			
<20 years	1.00		
≥20 years	1.16	0.27, 4.93	0.840
**Alcohol**			
**Current**	1.48	1.12, 1.96	0.007 *
**Timing of drinking**			
<20 years	1.00		
≥20 years	0.82	0.48, 1.38	0.452
**Drinking frequency per week**			
<5 times	1.00		
≥5 times	1.19	0.70, 2.03	0.518
**Former**	1.10	0.72, 1.68	0.652
**Period of quit drinking**			
<10 years	1.00		
≥10 years	1.67	0.76, 3.68	0.206
**Period of past drinking**			
<10 years	1.00		
≥10 years	0.76	0.33, 1.78	0.529

Data were analyzed with simple logistic regression. * Significant level at *p* < 0.05.

**Table 4 ijerph-18-08738-t004:** Adjusted odds ratios of personal data, and risk factors with OPMD occurrence.

	Adjusted Odds Ratio	(95% CI)	*p*-Value
**Gender**			
Male	1.00		
Female	1.64	1.10, 2.43	0.015 *
**BQ**			
Chewer/former chewer	4.65	3.29, 6.58	<0.001 *
Non-chewer	1.00		
**Alcohol**			
Drinker/former drinker	3.40	2.23, 5.18	<0.001 *
Non-drinker	1.00		

Data were analyzed with multiple logistic regression using the forward likelihood ratio method. * Significant level at *p* < 0.05.

**Table 5 ijerph-18-08738-t005:** Odds ratio of combined risk factor exposure characteristics with OPMDs occurrence.

Risk Factors	PMD (N = 562)	No PMD (N = 886)	Odds Ratio	95% CI	*p*-Value
**1 risk factor**					
Tobacco	80 (14.2)	131 (14.8)	0.89	0.66, 1.21	0.467
SLT	64 (11.4)	28 (3.2)	3.98	2.52, 6.30	<0.001 *
BQ	346 (61.6)	180 (20.3)	6.91	5.46, 8.75	<0.001 *
Alcohol	110 (19.6)	126 (14.2)	1.48	1.12, 1.96	0.007 *
**2 risk factors**					
Tobacco + SLT	0 (0.0)	0 (0.0)	NA		
Tobacco + BQ	9 (1.6)	5 (0.6)	2.87	0.96, 8.60	0.060
Tobacco + Alcohol	37 (6.6)	51 (5.8)	1.15	0.75, 1.79	0.521
Tobacco + SS	0 (0.0)	0 (0.0)	NA		
SLT + BQ	57 (10.1)	21 (2.4)	4.65	2.79, 7.76	<0.001 *
SLT + Alcohol	17 (3.0)	2 (0.2)	13.79	3.17, 59.91	<0.001 *
SLT + SS	0 (0.0)	2 (0.2)	NA		
BQ + Alcohol	59 (10.5)	11 (1.2)	9.33	4.86, 17.93	<0.001 *
BQ + SS	77 (13.7)	45 (5.1)	2.97	2.02, 4.36	<0.001 *
Alcohol + SS	19 (3.4)	9 (1.0)	3.41	1.53, 7.59	0.003 *
**3 risk factors**					
Tobacco + BQ + Alcohol	5 (0.9)	0 (0.0)	NA		
SLT + BQ + Alcohol	13 (2.3)	1 (0.1)	20.96	2.73, 160.64	0.003 *
BQ + Alcohol + SS	18 (3.2)	4 (0.5)	7.30	2.46, 21.67	<0.001 *

Data were analyzed with simple logistic regression. * Significant level at *p* < 0.05. NA: no subjects for analysis.

**Table 6 ijerph-18-08738-t006:** Rates of OPMD occurrence with 10, 30, and 50 years of risk factor exposure.

CP (%) (95% CI)	Duration of Exposure			
	10 Years	30 Years	50 Years	Median (95% CI)
**Tobacco**	7.5 (3.8, 11.2)	28.2 (21.1, 35.3)	57.9 (47.9, 67.9)	50 (48.49, 51.51)
**SLT**	16.1 (6.1, 26.1)	50.6 (39.2, 62.0)	82.9 (72.7, 93.1)	30 (23.62, 36.38)
**BQ**	12.3 (9.4, 15.2)	44.7 (40.0, 49.4)	83.6 (79.5, 87.7)	40 (37.90, 42.09)
**Alcohol**	20.3 (14.8, 25.8)	49.8 (42.0, 57.6)	78.5 (68.7, 88.3)	36 (30.49, 41.52)

CP, cumulative proportion. Median of exposure duration compared by log-rank test.

## Data Availability

All the associated data are available within the manuscript.

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
