# Peer review of "The Relationship of Tobacco, Alcohol, and Betel Quid with the Formation of Oral Potentially Malignant Disorders: A Community-Based Study from Northeastern Thailand"

_ijerph, 2021, doi:10.3390/ijerph18168738_

Round 1

Reviewer 1 Report

I am pleased to evaluate the manuscript entitled “The Relationship of Tobacco, Alcohol, and Betel Quid with the Formation of Oral Potentially Malignant Disorders: A Community-based Study from Northeastern Thailand”.

It is interesting to analyze the association of risk factors such as smoking and drinking with the formation of Oral Potentially malignant Disorders. However, many errors were found in the analysis method and interpretation of the paper.

Introduction

- Some sentences do not have citation. Please provide the appropriate citation.

  • The aim of the study is not clear based on the Introduction. Please use one sentence for the study aim: “The aim of the study was to evaluate...”

Materials and Methods

  • Why is there a difference between the rationale for determining the sample size and the actual number of  participants? What was the inclusion and exclusion criteria?
  • Please add the actual number of participants per follow-up in "Diagram of project workflow". In particular, were there any dropouts during the follow-up period?

Results

  • Please combine Tables 3-6 into one table.
  • There is an error in the classification for age or period. For example, if you classify as <30 years old and >30 years old, subjects who are actually 30 years old are excluded.
  • Please describe the statistical analysis method described along with the table title at the bottom of the table.

Discussion

  • Please delete the subheadings for each variable in the review.
  • In addition, in the last paragraph of the Discussion, please describe the significance and limitations of this study in general, and describe future research.

Reviewer 2 Report

Dear Authors,

The article: "The Relationship of Tobacco, Alcohol, and Betel Quid with the Formation of Oral Potentially Malignant Disorders: A Community-based Study from Northeastern Thailand"was to describe the relationship between main risk factors for oral squamous cell cancer, including tobacco (in form of cigarette, smokeless tobacco [SLT], secondhand smoking [SS]),
alcohol, and betel quid (BQ), and the occurrence of oral potentially malignant disorders (OPMDs).

English language and style are fine.

Authors should add information about aim of the study (should be clearly defined).

The materials and results must be inproved. Authors must add informations about inclusion and exclusion criteria.
Than, give years of research and provide the number of the relevant ethics committee.

Authors must add information on statistics at the end of the materials and methods section.
The country should be indicated next to the name of the statistical program.

There are too many tables in the results. Add some graphs results part.
Table 2 - There is an error in the classification for age. If you classify as <60 years old and >60 years old, subjects who are actually 60 years old are excluded.
The same mistakes in table 3, 4, 5 and 6: Timing of tobacco smoking (<30 and > 30); Smoking frequency per week (<35 and > 35); Period of quit smoking (<10 and > 10);
Period of past smoking (<20 and > 20).

I think table 3, 4, 5 and 6 should be connected.

p value should be written italics.

Discussion is clearly presented.

To sum up, article can be accepted after minor revison. 

Round 2

Reviewer 1 Report

I have confirmed that the revised manuscript entitled “The Relationship of Tobacco, Alcohol, and Betel Quid with the Formation of Oral Potentially Malignant Disorders: A Community-based Study from Northeastern Thailand” mostly reflected the review comments. However, there is an error in the added reference numbering system, so the citation number needs to be re-edited in order. (red)

  • Line 27: Although OPMDs are less commonly found in females[4,25,26,29]